# Clinical Applications of PSMA PET Examination in Patients with Prostate Cancer

**DOI:** 10.3390/cancers14153768

**Published:** 2022-08-02

**Authors:** Sazan Rasul, Alexander R. Haug

**Affiliations:** 1Department of Biomedical Imaging and Image-Guided Therapy, Division of Nuclear Medicine, Medical University of Vienna, 1090 Vienna, Austria; sazan.rasul@meduniwien.ac.at; 2Christian Doppler Laboratory for Applied Metabolomics (CDL AM), Medical University of Vienna, 1090 Vienna, Austria

**Keywords:** prostate cancer, diagnosis, PSMA, PSA, tumor, PET scan

## Abstract

**Simple Summary:**

The prostate specific membrane antigens, abbreviated as PSMAs, are type II membrane proteins that are highly ex-pressed on the surface of malignant prostate tissue in prostate cancer (PCa), particularly in aggressive, andro-gen-deprived, metastatic, and hormone-refractory PCa. Today, radionuclides that bind to these PSMA peptides are widely available for diagnostic and therapeutic purposes to specifically image and target prostate tumor cells at molec-ular level. In this descriptive review, we aimed to emphasize the usefulness of PSMA positron emission tomography (PET) examination in the management of patients with various stages of PCa. In addition, we outlined the main pitfalls and limitations of this scan to avoid misinterpretation of the results and to improve the decision making process in rela-tion to the patient’s further treatment. We concluded that PSMA PET examination in primary PCa patients has an es-sential role in the high-risk group. It is the new imaging standard in patients with in biochemical recurrence PCa and plays an important role in treatment decision. Furthermore, PSMA PET scan is a gold standard for the evaluation of PSMA targeted therapies in patients having progress of the disease. Future prospective studies, particularly on the im-pact of PSMA PET on therapy stratification, may further strengthen the role of PSMA in the treatment of PCa patients.

**Abstract:**

With the progressive aging of the population in industrially developed countries, as well as advances in diagnostic and biopsy techniques and improvements in patient awareness, the incidence of prostate cancer (PCa) is continuously increasing worldwide. Therefore, PCa is currently considered as the second leading cause of tumor-related death. Early detection of the tumor and its metastasis is essential, as the rate of disease recurrence is high and occurs in 27% to 53% of all patients who underwent curative therapy with radical prostatectomy or local radiotherapy. In this regard, the prostate specific membrane antigens, abbreviated as PSMAs, are type II membrane proteins that are highly expressed on the surface of malignant prostate tissue in PCa, particularly in aggressive, androgen-deprived, metastatic, and hormone-refractory PCa, and they are inversely associated with the androgen level. Up to 95% of adenocarcinomas of the prostate express PSMA receptors on their surface. Today, radionuclides that bind to these PSMA peptides are widely accepted for diagnostic and therapeutic purposes to specifically image and target prostate tumor cells at the molecular level, a process referred to as targeted theranostics. Numerous studies have demonstrated that the integration of these peptides into diagnostic and therapeutic procedures plays a critical role in the primary staging and treatment decisions of especially high-risk PCa, expands therapeutic options for patients with advanced stage of prostate tumor, and prolongs patients’ survival rate. In this review article, we intend to briefly spotlight the latest clinical utilization of the PSMA-targeted radioligand PET imaging modality in patients with different stages of PCa. Furthermore, limitations and pitfalls of this diagnostic technique are presented.

## 1. Introduction

Prostate cancer (PCa) remains one of the primary causes of tumor-related deaths in men. It is estimated that, after lung cancer, PCa is the second leading cause of cancer death, with the average age of men at time of diagnosis being about 66 years. Since the 5 year survival rate of PCa is highly dependent on tumor stage (ranging from 100% in early-stage tumors to as low as 30% in patients with advanced cancer), early detection and accurate staging of the disease are essential [1,2]. Prostate-specific antigen (PSA) that is produced by the epithelial cells of the prostate predicts prostate cancer and provides an important indicator of tumor recurrence after primary therapy for PCa [3]. Although surgical radical prostatectomy (RP) is the curative therapy of PCa, up to 50% of all patients might develop an increase in PSA in terms of biochemical recurrence (BCR) after 5 years. At this clinical stage, systemic androgen deprivation therapies (ADT) may be considered the treatment of choice in most patients. In addition to the serious side-effects of this therapy, such as hot flashes, impotence and sexual dysfunction, metabolic syndrome, bone loss, and increased risk of cardiovascular disease, which negatively affect the quality of life of these patients, patients develop castration resistance after a certain period of therapy. 

Prostate-specific membrane antigen, which is abbreviated as PSMA and called glutamate carboxypeptidase type II, is a type II membrane protein, composed of collectively 750 amino acids and located on chromosome 11. Its level increasingly expressed of the surface of the inflammatory, benign, and malignant prostatic tissues. However, the level of this antigen increases up to thousand times its normal level in the case of prostate cancer (PCa), especially in the case of aggressive types with high Gleason scores (GS), as well as in androgen-deprived, metastatic, and hormone-refractory PCa [4].

PSMA was discovered in late 1980s and. since then, it has become an attractive target for the diagnosis and therapy of prostate cancer and its associated metastasis. Therefore, it has become the focus of numerous therapeutic approaches such as PSMA-based immunotherapies, including anti-PSMA antibodies and PSMA-targeted prodrug treatments. In the early 2000s, antibodies directed against the cytoplasmic part of the PSMA antigen were labeled with [^111^indium] and used for the first time for the scintigraphical imaging [5]. However, to improve image quality and reduce radiation exposure for the patients, the researchers at Heidelberg University were able to develop in 2012 for the first time a [^68^Ga]Gallium-labeled PSMA ligand suitable for positron emission tomography (PET) scanning that targets the extracellular component of PSMA [6]. In the General Hospital of Vienna, Medical University of Vienna, Austria, we could perform the first [^68^Ga]Gallium PSMA PET examination for patients with PCa in May 2013. Since then, we have conducted more than 5000 PSMA PET examinations for patients with primary PCa, as well for patients with biochemical recurrent PCa and patients with metastatic hormone-sensitive (mHSPC) and castration-resistant PCa (mCRPC) [7,8,9,10].

Meanwhile, there are many available PSMA ligands labeled either with [^68^Ga]Gallium or [^18^F]Fluoride such as [^68^Ga]Ga-PSMA-11, [^68^Ga]Ga-PSMA-I&T, [^68^Ga]Ga-PSMA-11 gozetotide, [^18^F]F-PSMA-1007, [^18^F]F-DCFPyL, [^18^F]rh-PSMA7, [^18^F]JK-PSMA-7, and [^68^Ga]Ga-PSMAR2. Until now, the United States FDA has only approved [^18^F]F-DCFPyL and [^68^Ga]Ga-PSMA-11 gozetotide for clinical use, and they are, therefore, the only PSMA PET tracers that are currently commercially available. The approval of these tracers via FDA was based on clinical outcomes of the phase II/III multicenter Osprey study [11,12], as well as Condor [13] and Vision trials [14], which could demonstrate the safety and the diagnostic performance of these tracers in patients with locally advanced PCa, as well as in patients with metastatic and biochemical recurrent tumor.

In this brief review, we aimed to focus on the current clinically relevant indications of PSMA PET examinations in patients with various stages of PCa. In addition, we aimed to outline the main pitfalls and limitations of this scan to improve the diagnostic and therapeutic management of patients with PCa.

## 2. Clinical Report of the PSMA PET Scan

Before writing the clinical report and interpreting the PSMA PET scan results, it is important to know that PSMA is not specific to prostate tissue and is physiologically expressed on the cellular surface of other organs such as the lacrimal, parotid, and salivary glands as well as brain, proximal tubule of kidneys, small intestine, liver, and spleen [15]. This physiological expression of PSMA in these organs varies from organ to organ, with the highest PSMA expression in the kidneys and salivary glands and the lowest in brain tissue. However, the intense upregulation of this receptor beyond its normal level results in lesions and metastases associated with PCa having visually and quantitatively much higher PSMA tracer avidity on PET scan; this greatly facilitates their differentiation from background and neighboring organs physiologically expressing PSMA. Furthermore, the presence of computed tomography (CT) and magnetic resonance imaging (MRI) as a part of the integrated PET examination might sometimes play a critical role in the morphological confirmation of these lesions. Nevertheless, PSMA PET scans in patients with PCa should be carefully assessed to avoid pitfalls and misinterpretation of results and findings. After all, this has a direct impact on further steps in the therapy and management of these patients. Moreover, the biodistribution of all abovementioned PSMA PET tracers are almost similar except for [^18^F]F-PSMA-1007, as this tracer is almost exclusively excreted via the hepatobiliary duct [16].

For lesion evaluation, the European Association of Nuclear Medicine recommends visual and quantitative evaluation of the lesions expressing PSMA. Visual evaluation involves determining the PSMA expression level of the lesion compared to background, also called PSMA expression V. Quantitative evaluation involves measuring the standard uptake values (SUV) of the focal tracer uptake and is referred to as PSMA expression Q [16]. This recommendation was based on the results of the PROMISE study that divided PSMA expression into four scores. Score 0 is when there is no PSMA expression or expression below the blood pool. Score 1 is when expression is low or equal to or above the blood pool and lower than the liver. Score 2 is intermediate with expression equal to or above the liver and lower than the parotid gland. Score 3 is high with expression equal to or above the parotid gland [16]. Lesions with scores of 2 and 3 are considered very typical for prostate cancer-related lesions, and they are favorable for PSMA-directed radioligand therapy.

## 3. Clinical Indications of PSMA PET Examinations in Prostate Cancer

The most clinically relevant indications for PSMA PET examinations in patients with PCa are in the primary staging of the tumor, in BCR and castration-resistant PCa, and in the evaluation for PSMA targeted therapies. However, it is important to know that up to 5% of prostate adenocarcinomas do not express PSMA [17,18].

### 3.1. Roles of PSMA PET in Primary Staging of Prostate Cancer

In this setting, although studies showed that the primary tumor is nearly always detected by PSMA PET reaching a sensitivity up to 99.3%, and that the PET metrics correlate with histology grades and International Society of Urological Pathology (ISUP) classifications and GS, high variation in sensitivity (between 33% to 92%) with a very good specificity of 82% to 100% was found in the detection of lymph node metastases [7,19]. Current guidelines of European Association of urology recommend this examination only for high-risk PCa patients, i.e., patients who have an initial PSA level ≥ 20 ng/mL, a pathologically ISUP grade of 3 to 5, or a clinical stage of T3 or greater at the time of PCa diagnosis, which means the tumor has exceeded the prostate capsule. In this context and in relation to local grading of the tumor, studies have compared integrated PSMA PET-MRI with multiparametric (mp)MRI in patients with primary PCa. These studies showed that integrated PET-MRI has superior diagnostic accuracy compared with mpMRI and is particularly useful in tumors with equivocal Prostate Imaging Reporting and Data System (PIRADS) classification of 3 [20,21,22]. Recently, we were able to confirm these results in a meta-analysis of five prospective studies with collectively 497 men and could highlight the favorable diagnostic precision of PSMA PET targeted biopsy to detect clinically significant PCa [23]. Furthermore, local characteristics of the tumor such as tumor infiltration into the seminal vesicle, as well as into the other neighbor organs like the urinary balder, rectum, and neurovascular bundle, can exactly be determined using combined PET-MR examination [7]. Muehlematter et al. could illustrate in a retrospective assessment of 40 consecutive men who performed mpMRI and PSMA PET-MRI followed by a RP due to intermediate- to high-risk PCa a better sensitivity of PSMA-PET/MRI to detect extracapsular extension and infiltration of the seminal vesicle than with mpMRI [24]. Actually, despite the limited availability of comparative head-to-head studies with [^18^F]F-PSMA-1007 and [^68^Ga]Ga-PSMA-11 for prostate tumor staging, several studies have reported that both these tracers can equally identify predominant prostate lesions in patients with intermediate- or high-risk PCa, although the nonurinary excretion of [^18^F]F-PSMA-1007 may help evaluate lesions near the urinary bladder better than [^68^Ga]Ga-PSMA-11 [25,26,27].

Concerning the primary staging of the disease, results of the proPSMA study were recently published in Lancet (2020), which was a prospective multicenter two-arm randomized controlled trial including more than 300 patients with histologically confirmed high-risk PCa being considered for curative therapies [28]. One arm received CT and bone scans for the primary staging of their tumor, and the other arm received only PSMA PET scan. Results showed that PSMA PET staging is a suitable replacement for conventional imaging, providing superior accuracy to the combined findings of CT and bone scans, a higher proportion of management changes, and fewer equivocal results. In addition, the radiation exposure of the patients was significantly lower with PSMA PET scan than with the CT and bone scan. Concerning this topic and modifications in the therapy management of patients with primary PCa, Figure 1 presents the PSMA PET-CT images of a 56 year old man, newly diagnosed with PCa: GS (5 + 5) and a PSA value of 45.96 ng/mL at the time of diagnosis. The patient was scheduled for RP and extended pelvic lymphadenectomy. However, the PSMA-PET-CT examination prior to his surgical approach could detect two PSMA-avid bone lesions (frontoparietal in the left side of the skull and in the ninth thoracic vertebra left side), in addition to the primary tumor in the prostate and bilateral iliac lymph node metastases (Figure 1). Both these bone lesions were not visible on CT examination separately performed for the patient. The patient was then treated systemically with hormonal and chemotherapy instead of the primary surgical intent.

### 3.2. PSMA PET in Biochemical Recurrence PCa

Current guidelines of the European Association of Urology suggest a PSMA PET scan in any case of proven BCR, as well as in patients where the results of the examination will influence the subsequent treatment decisions. Indeed, the cutoff value for the definition of a PSA recurrence, which usually suggests the relapse of the disease, depends strongly on the type of primary tumor treatment. European Association of Urology defines a BCR of PCa as an increase in PSA to at least 0.2 ng/mL after RP with a cutoff value above 0.4 ng/mL most likely to predict further metastases. However, in patients treated primarily with local radiotherapy with or without hormonal therapy, a rise in PSA level of 2 ng/mL or more above the nadir level is determined as BCR based on RTOG-ASTRO Phoenix Consensus Conference in 2006 [29]. In this regard, identification of lesions responsible for the recurrence of the PCa in such a patient cohort, however, could sometimes be like searching for a needle in a haystack, especially in patients with very low PSA recurrence levels. Considering the role of imaging, conventional imaging with bone scan provides very low sensitivity for patients with a PSA level below 2 ng/mL, and positive bone scintigraphy can only be identified in approximately 9.4% of patients at that PSA level. Furthermore, findings indicated that the ability of CT scan to detect biochemical recurrence is only 11–14%. Kane et al., in a retrospective analysis of 132 men with biochemical relapse after RP, displayed a mean PSA level of 27.4 ng/mL, and a PSA velocity of 1.8 ng/mL/month was associated with a positive CT result [30]. Although [^18^F]choline and [^18^F]fluciclovine PET examinations have higher sensitivity (up to 90%) and specificity (up to 100%) than bone scans for finding bone metastases in patients with BCR [31,32], sensitivity of both these tracers is highly dependent on the PSA level and is less than 50% at PSA lower than 1.0 ng/mL [33], with poorer sensitivity of [^18^F]choline for detecting lymph node metastases than [^18^F]fluciclovine PET scan. However, if we compare PSMA PET with both these tracers, we can see that prospective and retrospective studies could demonstrate significantly higher sensitivity and detection rate of PSMA (58%) than choline (35%) or fluciclovine (23%), especially in patients with a PSA level lower than 0.5 ng/mL [34,35]. Regardless of the PSA level, PSMA PET can also detect small, choline PET-negative lymph node metastasis as could be demonstrated in previous studies, which showed PSMA-positive lymph nodes having diameters of only 6 mm [34]. Furthermore, PSMA PET offers not only higher sensitivity than choline PET but also higher contrast and higher specificity [34,36].

Furthermore, the presence and the timepoint of certain treatments, such as ADT, chemotherapy, and immunotherapy, could have an important influence on the sensitivity and specificity of the PSMA PET scan [37]. Although both in vivo and in vitro studies [38,39,40] have shown that ADT upregulates PSMA expression, with enhanced PSMA expression demonstrated in 55% of patients with high-grade and advanced PCa treated with ADT, it appears that PSMA expression rises with a short duration of ADT and declines with a longer duration of therapy [41,42]. Thus, it is not yet clear exactly how ADT affects the diagnostic value of this scan, necessitating further potential clinical prospective studies to address this issue, since the duration of ADT represents a critical factor.

In fact, the detection rate of PSMA PET increases with higher PSA levels and higher GS as can be seen in a study that included more than 200 patients with PCa. The sensitivity of PSMA PET imaging reached 46% even with a PSA level lower than 0.2 ng/mL [43]. This high sensitivity of PSMA to detect metastasis plays an important role in the treatment decision of prostate cancer patients. In a large meta-analysis that included 15 studies with more than 1100 patients mainly with BCR but also with primary PCa, the high impact of PSMA on changing the therapy and management of patients was demonstrated. With the help of the PSMA PET imaging, patients received more surgery and more radiation therapy than systemic treatments with hormonal or chemotherapies [44].

In this setting, it is further important to mention the study by Jentjens et al. [45] that included 34 patients with BCR, which prospectively underwent a PSMA PET-CT followed by PSMA PET-MRI scan. Results revealed significantly higher sensitivity of PET-CT and PET-MRI compared to CT and MRI. Although PET-MRI could detect more lesions than PET-CT, results indicated no significant differences between these two modalities in the detection of the presence of local recurrence or distant metastases. One other study could indicate the superiority of PET-MRI over PET-CT in the identification of local PCa recurrence in 119 patients with BCR due to the additional diagnostic information derived from MRI [46].

### 3.3. Evaluation of PCa Patients for Possibility of PSMA Radioligand Therapies

Another important clinical indication of PSMA PET examination in patients with PCa is to reveal whether their tumor-related metastases express PSMA for the purpose of potential PSMA radioligand therapies. The most common radionuclide for therapy is [^177^Lu]Lutetium-labeled PSMA-617 ligand, which is generally abbreviated as PSMA-RLT. Although the rate of response to this therapy varies from 60–75% [47,48], treatment options for mCRPC patients have expanded since the introduction of this therapy, and numerous studies have verified the antitumor efficacy, as well as the good tolerability and favorable response rates, of PSMA-RLT [49,50,51,52]. However, it is currently used mainly as a last-line treatment when other available standard therapies fail, and the therapeutic protocols are very heterogeneous in terms of different treatment cycles, as well as different therapy doses and different inter-cycle intervals, depending on the general medical status of the patients. Multiple baseline characteristics such as the distribution of metastases, history of pretreatment with hormonal and chemotherapies, number and interval between PSMA-RLT, and the baseline serum levels of PSA, platelets, hemoglobin, alkaline phosphatase, and lactate dehydrogenase have been found to play an essential role in helping to anticipate the response to PSMA-RLT [47,53,54,55,56]. In addition, identification of the metabolic activity of the metastatic tumors using [^18^F]fluorodeoxyglucose (FDG) aids in detecting the aggressive nature of these lesions [57] and allows prediction of their response to PSMA-RLT prior to therapy; lesions that are [^18^F]FDG-positive but PSMA-negative on PET examination represent more aggressive and metabolically active lesions with worse outcome and response to PSMA-RLT than PSMA-positive but FDG-negative ones [58,59].

In this context, two significant studies in the field of PSMA-RLT are worth mentioning. One is the TheraP study, which was a multicenter, unblinded randomized phase II trial, involving 11 centers in Australia and including about 200 mCRPC patients; 98 of these patients were treated with the PSMA-RLT and 85 were treated with cabazitaxel chemotherapy at the same stage of the disease [60]. Results of the study revealed that patients who received PSMA-RLT hade a significantly higher percentage of PSA response and fewer serious adverse events than patients receiving cabazitaxel. Another important study is the VISION study sponsored by Novartis, which was an international prospective randomized open-label multicenter phase III study [14]. The study included two groups of patients with mCRPC; one group received PSMA RLT plus best standard of care (551) compared to a group of patients that received only best standard of care in the control arm. In this trial, all patients with one and/or more PSMA-negative lesions on PSMA PET scan were excluded from the study. Results showed that men in the investigational arm had a 38% reduction in risk of death and a 60% reduction in risk of radiographic disease progression or death compared to men in the control group, with a significant improvement in time to first symptomatic skeletal event, overall response rate, and disease control rate. Notably, the VISION study had 20% fewer patients with PSA reductions greater than 50% than the TheraP trial. Factors that might contribute to these results were the higher exclusion rate of patients in the TheraP study compared with the VISION study, as the study excluded all patients with FDG-positive but PSMA-negative lesions on their PET scan, and the stricter selection criteria in the TheraP study than in the VISION study, as only patients with lesions with a maximum standardized PSMA uptake value of ≥20 were enrolled.

Figure 2 is an example of the role of PSMA PET examination in a patient with mCRPC, who had a PSA level of 65.74 ng/m before PSMA-RLT. After receiving six cycles of this therapy, the PSA level declined to 1.15 mg/mL, and the metastases largely disappeared on the PSMA PET scan.

## 4. Pitfalls and Limitation of PSMA PET Examination

As mentioned earlier, PSMA is not specific to prostate tissue, and physiologically PSMA-expressing areas or organs (lacrimal glands, parotid and salivary glands, brain, and proximal tubules of kidneys, small intestine, liver, and spleen) should be well distinguished from lesions that pathologically express PSMA. Furthermore, PSMA radioligand and especially [^68^Ga]Ga-PSMA will be excreted via kidney and urinary bladder. Thus, an accumulation of this tracer throughout the urinary tract or in a specific segment of the ureter might be misinterpreted as a suspicious lymph node. Furthermore, [^68^Ga]Ga-PSMA ligand uptake in sympathetic ganglia such as cervical, celiac, and sacral ganglia is very common should be carefully taken into consideration. These focal uptakes, however, are usually symmetrical and bilateral, and they could radiologically be differentiated from lymph nodes [61]. Moreover, some comparative studies demonstrated identification of nonspecific bone lesions with [^18^F]PSMA-1007 but not with other PSMA PET tracers in patients with PCa [62,63].

Figure 3 displays the case of a 74 year old PCa patient treated with RP in 2011 with initial PSA of 14 ng/mL and GS 7 (3 + 4). After prostatectomy, the patient received no further therapies. Seven years later, the patient developed BCR with a PSA level of 0.7 ng/mL; a few months later, the PSA reached 1.22 ng/mL with a PSA doubling time of about 12 months. A PSMA PET examination was performed for the patient. The scan showed PSMA-positive bone lesions in the left iliac bone and in the vertebral arch of the fourth lumbar spine. In addition, PSMA-positive intrahepatic lesions were detected in liver segments IVa and IVb. However, no local recurrence and no suspicious lymph nodes were found. This case was discussed by our institutional tumor board, and a histological clarification of these intrahepatic PSMA-avid lesions was strongly recommended as PSA levels and the extent of liver metastases did not match well. Histopathology results demonstrated hepatocellular carcinoma rather than prostate cancer metastases in these lesions.

Indeed, expression of PSMA is not specific to prostate tissue and PCa, as it might be increasingly expressed in the process of neovascularization of numerous solid tumors such as renal cell cancer [64], hepatocellular carcinoma, and squamous cell carcinoma of head and neck [65] (see Figure 4). In a prospective pilot study by Kessler et al. including 37 suspected malignant hepatic lesions of seven patients, results revealed that more than 95% hepatocellular carcinoma lesions expressed PSMA [66]. Consequently, in cases of low PSA levels and large PSMA-expressing organ lesions, possible etiologies other than PCa should be considered for the lesion. In this sense, in a review publication by Osmany et al. [67], the authors could show a list of benign and malignant conditions that might be associated with the increased expression of PSMA, including hepatic hemangioma, fractures, neuroendocrine tumor, lymphoma, multiple myeloma, osteosarcoma, breast cancer, and adenocarcinoma of the lung.

## 5. Conclusions and Outlooks

This review, which is descriptive rather than systematic, emphasized the usefulness of PSMA PET examination in the management of patients with various stages of PCa. It should provide clinicians with a concise overview of the clinical potential of this diagnostic imaging modality in patients with PCa. In addition, it was intended to help ascertain the pitfalls and vulnerabilities associated with this scan to avoid misinterpretation of the results and to facilitate the decision making process in relation to the patient’s further treatment.

We concluded that PSMA PET examination in PCa patients has an essential role in the high-risk group. It is the new gold standard in patients with BCR and plays an important role in treatment decision and selection of possible therapy options such as surgery, radiation, hormone, or chemotherapy treatment. Additionally, PSMA PET is a gold standard for the evaluation of PSMA targeted therapies in patients having progress of the disease after obtaining other available standard therapies.

Future prospective studies, particularly on the impact of PSMA PET on therapy stratification, may further strengthen the role of PSMA in the treatment of PCa patients. Nevertheless, the main issue with this investigation lies in its limited availability and reimbursement in hospitals and clinical centers, which reduce patients’ access to this valuable diagnostic imaging modality.

## Figures and Tables

**Figure 1 cancers-14-03768-f001:**
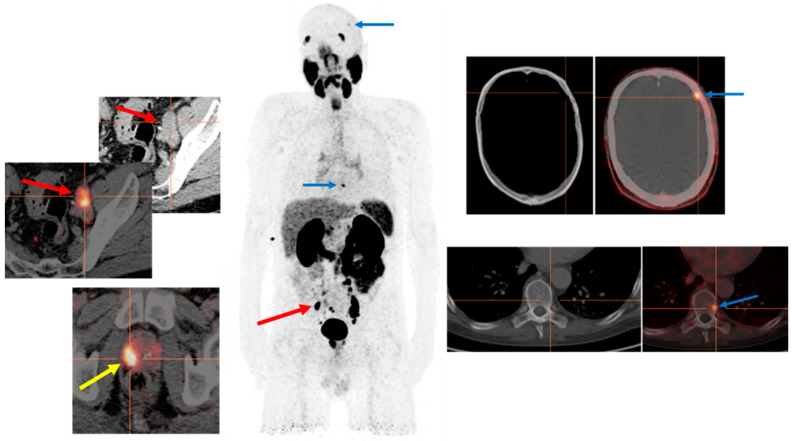
PSMA PET examination for primary staging of a high-risk prostate cancer. PSMA PET-CT Images for primary staging of a 56 year old man, newly diagnosed with PCa: Gleason score (5 + 5) and a PSA value of 45.96 ng/mL showing two PSMA-avid bone lesions frontoparietal in the left side of the skull and the ninth thoracic vertebra (blue arrows) in addition to the primary tumor in the right lobe of the prostate (yellow arrow) and bilateral iliac lymph node metastases (red arrows). Both these bone lesions were not visible on CT examination separately performed for the patient.

**Figure 2 cancers-14-03768-f002:**
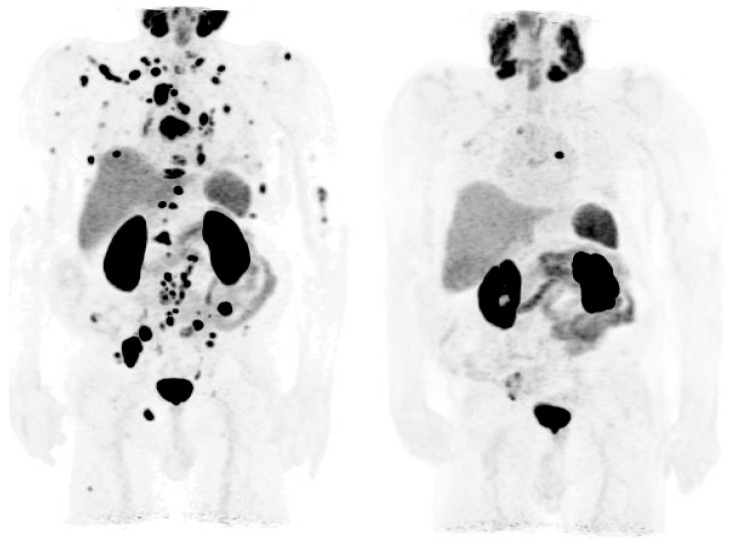
PSMA PET examination for evaluation of PSMA radioligand therapy. PSMA PET examination of a patient with metastatic castration-resistant prostate cancer and a PSA level of 65.74 ng/mL. Before PSMA-RLT (**left**), showing multiple bone and lymph node metastasis with highly increased PSMA-expression. After six cycles of PSMA-RLT and a PSA decline to only 1.15 ng/mL (**right**), showing a clear reduction in the number of PSMA-expressing bone and lymph node metastases.

**Figure 3 cancers-14-03768-f003:**
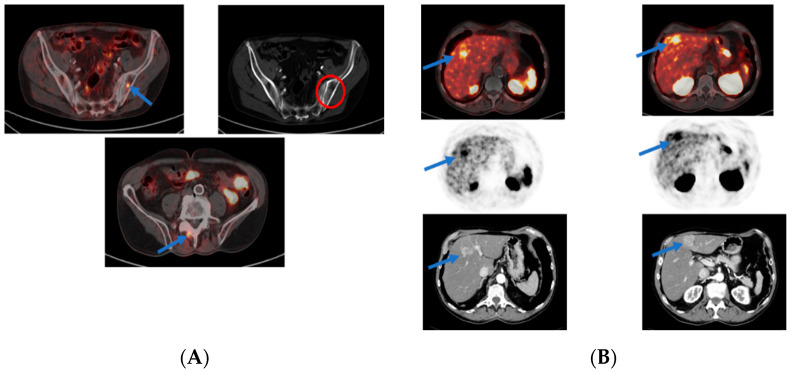
PSMA PET examination for biochemical recurrence of prostate cancer. [^68^Ga]Ga-PSMA PET-CT of a 74 year old patient with biochemical recurrence of prostate cancer revealed (**A**) PSMA-positive bone lesions in the left iliac bone with corresponding bone changes on the CT scan (red circle) and the vertebral arch of the fourth lumbar spine. (**B**) Additional PSMA-avid and contrast-enhanced intrahepatic lesions in liver segments IVa and IVb. Histological examination of these hepatic lesions demonstrated hepatocellular carcinoma.

**Figure 4 cancers-14-03768-f004:**
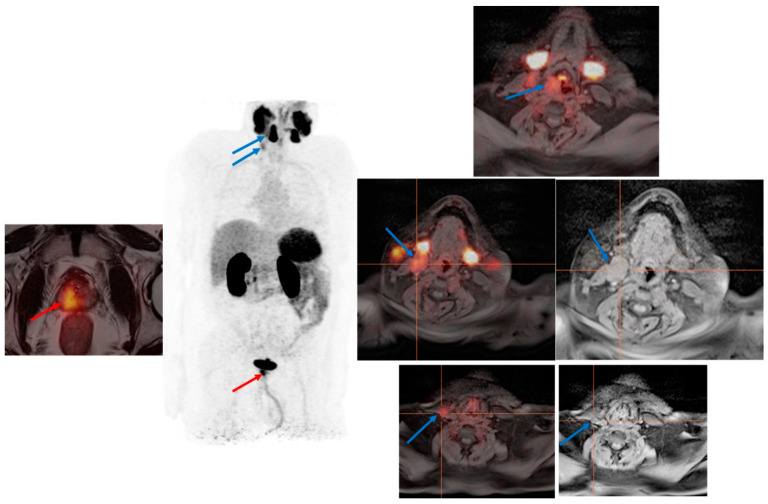
PSMA PET examination for primary staging of a high-risk prostate cancer. PSMA PET examination for primary staging of a 60 year old patient with a newly diagnosed high-risk prostate cancer (PSA value: 30.9 ng/mL) revealed the primary tumor in the right lobe of the prostate with a clear PSMA-expression (red arrow) and an incidental detection of a suspicious lesion with marked PSMA expression in the right piriform sinus (area of the right vocal cord) with multiple suspicious lymph nodes in the right neck region (blue arrows). The final histological examination of this neck lesion confirmed squamous cell carcinoma of the larynx.

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
