# Peer review of "Clinical Applications of PSMA PET Examination in Patients with Prostate Cancer"

_cancers, 2022, doi:10.3390/cancers14153768_

Round 1

Reviewer 1 Report

This is a well written manuscript. only one concern I have with the method is as follows. The authors says that PSMA can be expressed in lacrimal, parotid and salivary glands, brain, proximal tubule of kidneys, small intestine, liver and spleen. If there is metastasis to these organs, how will we determine if PSMA is coming from the metastatic prostate cells or local PSMA is being labeled and scanned? This Issue needs to be addressed properly. 

Author Response

Thank you for the positive evaluation of the paper and your important comments which have helped to improve the manuscript.

Main comments
This is a well written manuscript. only one concern I have with the method is as follows. The authors says that PSMA can be expressed in lacrimal, parotid and salivary glands, brain, proximal tubule of kidneys, small intestine, liver and spleen. If there is metastasis to these organs, how will we determine if PSMA is coming from the metastatic prostate cells or local PSMA is being labeled and scanned? This Issue needs to be addressed properly.

We appreciate your concern, as the physiological expression of PSMA on the cell surface of other tissues could affect the clinical interpretation of PSMA PET scanning in patients with prostate cancer, even though PSMA expression varies from organ to organ, with the highest PSMA concentration found in the kidneys and salivary gland and the lowest in brain tissue. However, in case of prostate cancer, the level of this antigen, as mentioned in the manuscript, will be upregulated to thousand times its normal level. Therefore, the prostate cancer related lesions and metastases can be differentiated from background and their neighbored organ physiologically express PSMA, as they exhibit quantitatively and visually much higher PSMA avidity. Furthermore, the presence of computed tomography (CT) and magnetic resonance imaging (MRI) in the setting of the integrated positron emission tomography could sometimes play a critical role in the morphological correlation of these lesions. Nevertheless, PSMA scans in patients with prostate cancer should be carefully assessed to avoid pitfalls and misinterpretation of results and findings. After all, this has a direct impact on the further steps in the therapy and management of these patients. We added this information to the “Clinical report of the PSMA PET scan” part of the paper and marked it red.

Reviewer 2 Report

The authors of “Clinical Applications of PSMA PET Examination in Patients 2 with Prostate Cancer” present a well written review of PSMA PET in relation to clinical scoring, primary staging of prostate cancer, PSAME PET for biochemical recurrence, and evaluation of targeted PSMA radionuclide therapy.  This review also includes a few clinical examples of PSMA PET.  While each of these topics is mentioned, there actual information presented is very cursory and does not provide the reader with much information besides a superficial mention of each application.

For example, the authors mention the various available PMSA PET radiopharmaceuticals but do not go into much detail about the differences in the imaging profile of each agent, including differences in clinical interpretation, other than PMSA-1007 which undergoes predominately hepatobiliary excretion.  Additionally, while the authors note that patients with PSMA avid disease may be candidates for PSMA radionuclide therapy, they do not discuss discordant hypermetabolic disease nor other potential confounders that may limit the utility of PSMA radionuclide therapy.

Specific questions that the authors may choose to address under each section:

Primary staging of prostate cancer: 

PSMA PET, similar to fluiciclovine and choline PET, has the highest false negative results for patients with an intact prostate.  Discussion should be made of PSMA PET vs MRI as well as PET-MRI for patients with primary staging.

• Discussion as to how different PSMA PET agents compare for disease in the prostate bed, especially compared to PSMA-1007.

PSMA PET in biochemical recurrence PCa

• There are multiple definitions of biochemical recurrence and the other definitions should also be detailed.

  Would like a more complete discussion as to the sensitivity/specificity per PSMA PET radiopharmaceutical and also helpful to describe organ specific accuracies (e.g. local, lymph node, bone).

No discussion as to how ADT timing, chemotherapy, immunotherapy affect PET sensitivity/specificity.

Evaluation of PCa patients for possibility of PSMA radioligand therapies:

This section does not really discuss the PSMA PET aspects of patient selection.  Namely, it does not discuss if PSMA PET is truly necessary for PSMA-RLT (Vision enrolled 85% of patients).  No discussion as to discordant hypermetabolic disease which is found consistently in large minority of populations and has a profound negative prognostic indication for PSMA-RLT. 

Reviewer 3 Report

The introduction gives sufficient information on the discussed topic and provides an excellent historical background to enable readers to better understand the problem identified by the authors.

However, I have a few comments and observations that require clarification:

The figures included require a detailed description, numbering, with a precise explanation of what they represent.

Moreover, the authors did not sufficiently discuss the implications of the research in the discussion and did not comment on the strengths and limitations of the manuscript. I found the discussion superficial. It is strongly recommended to provide a future perspective and conclusions after the discussion.

Round 2

Reviewer 2 Report

Edits made to the original article are good and provide a better review of PSMA PET and interpretation.